# Prenatal High-Fat Diet Combined with Microplastic Exposure Induces Liver Injury via Oxidative Stress in Male Pups

**DOI:** 10.3390/ijms241713457

**Published:** 2023-08-30

**Authors:** Mao-Meng Tiao, Jiunn-Ming Sheen, I-Chun Lin, Madalitso Khwepeya, Hong-Ren Yu

**Affiliations:** 1Department of Pediatrics, Kaohsiung Chang Gung Memorial Hospital, Chang Gung University College of Medicine, Kaohsiung 833, Taiwan; tmm@cgmh.org.tw (M.-M.T.); e5724@cgmh.org.tw (J.-M.S.); uc22@cgmh.org.tw (I.-C.L.); 2Chang Gung Medical Education Research Centre, Chang Gung Memorial Hospital, Linkou, Taoyuan 333, Taiwan; khwepeyamada@yahoo.com

**Keywords:** maternal high-fat diet, microplastics, oxidative stress, pups, apoptosis

## Abstract

Prenatal high-fat diet (HFD) or exposure to microplastics can affect the accumulation of liver fat in offspring. We sought to determine the effects of maternal HFD intake and microplastic exposure on fatty liver injury through oxidative stress in pups. Pregnant female Sprague–Dawley rats were randomly divided into maternal HFD (experimental group) or normal control diet (NCD; control group) groups with or without microplastic exposure. As a result, the following groups were established: HFD-L (HFD + microplastics, 5 µm, 100 μg/L), HFD-H (HFD + microplastics, 5 µm, 1000 μg/L), NCD-L (NCD + microplastics, 5 µm, 100 μg/L), and NCD-H (NCD + microplastics, 5 µm, 1000 μg/L). The pups were sacrificed on postnatal day 7 (PD7). Liver histology revealed increased hepatic lipid accumulation in pups in the HFD-L and HFD-H groups compared to those in the HFD, NCD-L, NCD-H, and NCD groups on PD7. Similarly, liver TUNEL staining and cellular apoptosis were found to increase in pups in the HFD-L and HFD-H groups compared to those in the HFD, NCD-L, NCD-H, and NCD groups. The expression levels of malondialdehyde, a lipid peroxidation marker, were high in the HFD, HFD-L, and HFD-H groups; however, the highest expression was observed in the HFD-H group (*p* < 0.05). The levels of glutathione peroxidase, an antioxidant enzyme, decreased in the HFD, HFD-L, and HFD-H groups (*p* < 0.05). Overall, oxidative stress with cellular apoptosis plays a vital role in liver injury in offspring after maternal intake of HFD and exposure to microplastic; such findings may shed light on future therapeutic strategies.

## 1. Introduction

According to previous studies, diet changes can increase the risk of developing metabolic syndrome and nonalcoholic fatty liver disease (NAFLD) [1]. NAFLD can occur earlier in life due to the intake of a high-fat diet (HFD) during pregnancy, which can lead to impairment in the offspring that can be observed later in their adult life [2]. Such impairment can lead to liver fibrosis and cirrhosis, which may require liver transplantation [3,4]. Previous studies have shown that prenatal HFD or steroids can induce liver steatosis, which is closely associated with oxidative stress and acute or chronic inflammation [5,6,7,8].

The environmental factors that contribute to microplastic exposure include susceptibility to the sea, diet, salty air fresheners, cosmetics, and drinking water [9]. These factors are considered critical for the growth and development of children and are associated with the risk of adult NAFLD [10,11,12]. Microplastics can accumulate in the liver and gut of rat pups, leading to disturbances in lipid metabolism and oxidative stress [13]. Maternal exposure to polystyrene microplastics has also been found to cause metabolic disorders in rat pups [14]. According to Luo [14,15], the concentration of microplastics in children is similar to that in the normal environment (0.1–5 g of microplastics weekly/person). Inflammation and oxidative stress occur in the livers of offspring exposed to prenatal HFDs [5,6].

To date, prenatal HFD is the most known cause of altered insulin/carbohydrate metabolism, plasma corticosterone, disturbed leptin, inflammation, and oxidative stress in the livers of offspring [2,5,8]. However, microplastics cause hepatic cytotoxicity and disrupt ATP production, lipid metabolism, oxidative stress, and inflammation [13]. The mechanism by which microplastics interact with HFD remains unknown. Few studies have reported the combined effects of prenatal exposure to microplastics and chemical contaminants on liver toxicity in offspring [16]; however, studies on the combined effects of microplastics and HFD are lacking. This study sought to elucidate the levels of oxidative stress and inflammation in the livers of rat pups exposed to prenatal HFD and microplastics, with a special focus on the development of liver steatosis. This study also sought to elucidate the potential impact of maternal exposure to an HFD combined with microplastics on liver health programming in pups.

## 2. Results

### 2.1. Liver of Pups and Mother

Lipid accumulation was found to increase in the liver of pups in the HFD-L (high-fat diet + microplastics 5 μm, 100 μg/L) and HFD-H (high-fat diet + microplastics 5 μm, 1000 μg/L) groups compared to those in the HFD group. This result indicates that regardless of the concentration of microplastics in the HFD, lipid accumulation increases in the liver of pups. Compared to the NCD (normal control diet) group, lipid accumulation increased in the liver of pups in the NCD-L (normal control diet + microplastics 5 μm, 100 μg/L) and NCD-H (normal control diet + microplastics 5 μm, 1000 μg/L) groups (*p* < 0.05) (Figure 1a,b).

After female rats gave birth and were sacrificed, their liver weight was determined. Maternal liver weight was found to increase in both the HFD and HFD-L groups compared to the NCD group (*p* < 0.05) (Table 1). However, no significant changes were observed in the NCD-L and NCD-H groups compared to the NCD group. The body weight of pups in the NCD-L, NCD-H, HFD, HFD-L, and HFD-H groups increased after prenatal exposure to microplastics compared to that of pups in the NCD group (Table 1). Only the HFD-L group showed an increase in liver weight compared with the HFD and NCD groups (Table 1).

### 2.2. Ileum of Pups

The ileum length of pups was found to decrease in the NCD-H group compared to that in the NCD group. However, ileal length did not significantly decrease in the NCD-L group compared to that in the NCD group, suggesting that a low microplastic concentration did not have the same impact as a high concentration. In contrast, neither HFD-L nor HFD-H caused a decrease in pup ileal length compared to HFD, despite a decrease in the HFD-L and HFD-H groups compared to the NCD group (Figure 2a,b).

### 2.3. Apoptosis and Inflammation in the Liver of Pups

Based on caspase-3 expression, apoptosis increased in the liver of pups exposed to 1000 μg/L of prenatal microplastics, but not 100 μg/L of prenatal microplastics, regardless of the diet (NCD or HFD) (Figure 3a,b). Therefore, higher concentrations of prenatal microplastics (5 μm, 1000 μg/L) have a greater impact on apoptosis in the liver of pups than lower concentrations. Additionally, liver inflammation (IL-6) only increased in the HFD-H group compared to the HFD group and increased in the NCD-L, NCD-H, and HFD groups compared to the NCD group (Figure 3a,c). Based on IL-6 expression, liver inflammation increased in pups prenatally exposed to HFD-H but not to HFD-L, compared to HFD alone. 

Liver apoptosis (TUNEL staining) was found to increase in the HFD-L and HFD-H groups compared to the HFD group and increase in the NCD-L, NCD-H, and HFD groups compared to the NCD group (Figure 4a,b). 

### 2.4. Oxidative Stress in the Liver of Pups

Oxidative stress (malondialdehyde (MDA), a lipid peroxidation marker and indicator of oxidative stress) increased in the livers of pups in the HFD-H, HFD-L, and HFD groups compared to the NCD group (Figure 5a,b). This result indicates that HFD, regardless of the concentration of microplastics, led to an increase in MDA expression. In addition, MDA expression was elevated in the NCD-L and NCD-H groups compared to that in the NCD group (Figure 5a,b).

Western blot analysis showed that the expression of the liver antioxidant (glutathione peroxidase 1 (GPX1)) decreased in pups in the HFD-H, HFD-L, and HFD groups compared to that in pups in the NCD group (Figure 6a,b).

## 3. Discussion

Based on the findings of this study, prenatal exposure to microplastics in combination with an HFD led to additive increases in liver lipid accumulation in pups. Prenatal HFD and exposure to a high concentration of microplastics led to a decrease in the length of the ileum villi of pups compared to NCD. Liver apoptosis and inflammation increased in pups prenatally exposed to microplastics in the NCD-L group compared to those in the NCD group. Further, the HFD-H group showed an even greater increase in apoptosis than the HFD group. Oxidative stress was found to increase in both the microplastic-exposed NCD and HFD groups compared to the NCD alone group; however, no additive effects were observed upon combining HFD with microplastic exposure. Furthermore, liver antioxidant activity decreased in the HFD, HFD-L, and HFD-H groups compared with that in the NCD group.

Microplastics with a particle size range of 0.1–10 μm can penetrate all animal organs, including the placenta [17,18]. These microplastics can accumulate in various organs, such as the gonads, intestine, liver, and brain [19], leading to a reduction in reproductive output in animals [20]. According to Wang et al. [21], the liver is particularly vulnerable to exposure to microplastics for 8 weeks, which results in abnormal hepatic lipid metabolism upon ingestion. The presence of microplastics in the liver can disrupt lipid and energy metabolism through oxidative stress and accumulation of microplastics in the tissue [19,22]. Indeed, the accumulation of microplastics in the liver, kidney, and gut follows tissue-accumulation kinetics, and their distribution patterns are highly dependent on the particle size [13]. For example, after 4 weeks of exposure, the accumulation of 5 μm microplastics in the kidney and gut was found to be higher than that in the liver, while the accumulation of 20 μm microplastics displayed the opposite trend [13]. Maternal exposure to 5 μm microplastics has been demonstrated to increase the risk of metabolic disorders in pups and have a greater impact on their health [14]. In our study, prenatal exposure to 5 μm microplastics induced liver lipid accumulation in rats on PD7, and this accumulation was further enhanced when combined with an HFD. Moreover, higher concentrations of microplastics resulted in increased liver lipid accumulation.

Microplastics have been found in the gastrointestinal tract, including Peyer’s patches in animals [22] and the gastrointestinal tract in humans [13,23,24]. Studies have shown that combined exposure to 50 and 500 nm polystyrene particles can cause severe dysfunction of the mouse intestinal barrier, leading to increased permeability [25]. Prolonged exposure to microplastics for 21 d has been associated with gut damage, alterations in the gut metabolome and microbiome, inflammation, and oxidative stress [26]. In this study, the length of the ileal villi of pups was shortened after maternal exposure to high concentrations of microplastics. However, an additive effect on ileum length was not observed after prenatal exposure to HFD. This finding could be attributed to the different concentrations of microplastics used, as different concentrations can have different effects on the length of the ileum of pups.

According to prior evidence, exposure to microplastics can increase intracellular apoptosis through the activation of caspase-3 [27]. Larger microplastic particles are more likely to induce apoptotic cell death than smaller particles [27]. In addition, an overload of calcium ions (Ca^2+^) after microplastic exposure has been demonstrated to trigger hepatocyte death [28]. However, hydrogen sulfide has been found to reduce microplastic-induced hepatic apoptosis and inflammation by preventing the accumulation of reactive oxygen species [29]. In the current study, prenatal exposure to high concentrations of microplastics resulted in increased cellular apoptosis in the liver of pups. However, when combined with HFD, prenatal exposure to high concentrations of microplastics did not have additive effects on liver cellular apoptosis.

Lu et al. [22] reported the development of liver inflammation, oxidative stress, and lipid deposition in zebrafish after seven days of exposure to microplastics. Larger microplastic particles induce inflammatory responses or apoptotic cell death within 24 h of exposure [27]. Based on the present study, exposure of liver cells to higher concentrations of microplastics during prenatal development caused inflammation, even when combined with prenatal HFD.

Qiao et al. [26] found that microplastics can induce intestinal inflammation and oxidative stress. Similarly, Browne et al. [30] reported that ingested plastics could transfer pollutants and additives to animals, resulting in oxidative stress. Liang et al. [25] showed that exposure to polystyrene microplastics and nanoplastics for 24 h can cause intestinal barrier dysfunction, primarily through reactive oxygen species-mediated epithelial cell apoptosis. In our study, liver inflammation increased in pups prenatally exposed to microplastics alone or combined with HFD. Cellular oxidative stress in the livers of pups increased with both microplastic exposure and HFD; however, the effect of additive oxidative stress did not increase with the combination of microplastics and HFD.

### 3.1. What the Current Work Adds to the Existing Knowledge

Prenatal exposure to microplastics can cause liver steatosis and inflammation in pups. Previous studies have not investigated the effects of prenatal microplastic exposure on the liver or intestines of pups after maternal HFD intake. In this study, we found that inflammation in the liver of pups was additive, even when combined with maternal HFD. The apoptotic effects of prenatal microplastics on the livers of pups were observed, but only at high concentrations, with additive effects after maternal HFD intake. Additionally, the length of the ileal villi decreased in pups exposed to high concentrations of microplastics compared to those exposed to low concentrations.

### 3.2. Study Limitations

This study had several limitations that may affect the overall interpretation and understanding of the relationship between prenatal microplastic exposure and the health outcomes of pups. First, this study did not assess insulin sensitivity or resistance in pregnant mothers nor accounted for differences in sugar intake. Second, the collection of stool samples from pups on PD7 proved challenging, making it difficult to confirm the status of dysbiosis at this stage. Third, we did not evaluate liver injury after microplastic ingestion by female pups despite its very important contribution to the transgenerational effect mechanism. Further research is needed to address these limitations and provide a more comprehensive understanding of the effects of prenatal microplastic exposure, especially the combined effects of microplastics and maternal HFD.

## 4. Materials and Methods

### 4.1. Animals

The study was conducted at Kaohsiung Chang Gung Memorial Hospital Animal Experimental Center of in Taiwan and was approved by the Institutional Animal Care and Use Committee of the Hospital (No. 2021083001). Sprague–Dawley rats were used to study and housed in the animal facility. The rats were kept in a 12 h light/dark cycle with lights turned on at 7 a.m. The litters were checked on a daily basis at 10:00 a.m. The rats, approximately 7–8 weeks old, were divided into two groups. One group of female Sprague–Dawley rats was fed HFD, and the other was fed normal-chow diet (NCD) for a minimum of 8 weeks before mating. The HFD had a composition of 23 g/100 g protein, 35.5 g/100 g carbohydrate, and 35.8 g/100 g saturated fat (58 kcal% fat) from soybean oil and coconut oil, while the NCD had a composition of 19.2 g/100 g protein, 67.3 g/100 g carbohydrate, and 4.3 g/100 g saturated fat. The HFD and NCD were purchased from the Research Diet Company in the USA. The male rats consumed NCD except during mating.

Afterward, the female rats were allowed 24 h to mate with the male rats while consuming HFD and NCD, respectively, and were separated after that and housed individually in standard plastic cages. On the 14th day after mating and confirming pregnancy, pregnant female Sprague–Dawley rats were randomly divided into two groups: maternal HFD (N = 12) or NCD (N = 12) exposure paradigms until delivery. Few pregnant females in the HFD or NCD paradigms (N = 8; low concentration = 4, high concentration = 4) were further fed with microplastics since conception. The different groups included HFD-L: HFD + microplastics (5 μm, 100 μg/L), HFD-H: HFD + microplastics (5 μm, 1000 μg/L), NCD-L: NCD + microplastics (5 μm, 100 μg/L), and NCD-H: NCD + microplastics (5 μm, 1000 μg/L). We compared NCD with NCD + microplastics and HFD with HFD + microplastics. At 7 days old (PD7), the male pups were sacrificed for study.

### 4.2. Tissue Preparation

On PD7, rats were anesthetized with 25 mg/kg of Zoletil and 23 mg/kg of Xylazine. They were then perfused continuously with normal saline using a peristaltic pump. The liver and ileum were removed and placed on an ice plate immediately. The lumen of the small intestinal tissues was carefully cleaned using ice-cold phosphate-buffered saline solution (pH 7.4). The tissues were then embedded in paraffin and prepared as Swiss rolls for further study. Hematoxylin and eosin staining were performed to evaluate the villi length. The liver tissue for immunohistochemistry was cut into pieces and embedded in paraffin. The remaining liver and ileum samples were stored at −80 °C and dissected for future analysis.

### 4.3. Western Blot

As in our previous study, the liver tissue Western blot was prepared and examined following the procedures reported [6]. The expressions of cleaved-caspase 3 (#9661, Cell Signaling, Denver, MA, USA) at a dilution of 1:1000, IL-6 antibody (ab6672, Abcam; Cambridge, MA, USA) at a dilution of 1:1000, glutathione peroxidase 1 (GPX1) (ab22604, Abcam, Cambridge, MA, USA) at a dilution of 1:1000, Malondialdehyde (MDA) (ab27642, Abcam, Cambridge, MA, USA) at a dilution of 1:2000, and GAPDH (ab181602, Abcam, Cambridge, MA, USA) at a dilution of 1:5000 were examined.

### 4.4. Hematoxylin and Eosin (H&E) Staining

The liver and small intestines were dissected and were fixed in 4% paraformaldehyde at 4 °C overnight. Subsequently, they were dehydrated in ethanol with a gradient of (70%, 75%, 85%, 90%, 95%, and 100%), hyalinized in xylene, and embedded at 55 °C in paraffin wax. Sections of the liver and small intestine (4 μm thick) were then cut and stained using an H&E staining kit (ScyTek Laboratories, West Logan, WV, USA). The histologic lesions were observed using a Leica DMI-3000 microscope equipped with a digital camera. Additionally, we used ImageJ (Fiji version 1.8.0) to quantify liver lipid accumulation and the length of ileum villi [31]. The image was first converted to an 8-bit gray-scale image, which was black-white inverted so the lipid droplets would appear black. The black-white inverted image was then applied to an upper threshold of 20 out of 255 of the grayscale to remove inter-hepatocyte structures not indicating lipid droplet features, followed by particle analysis. All particles with a circularity between 0.5 and 1.0 and a diameter between 0.1 μm and 50 μm were counted. The area fraction (%) occupied was by the counted particles. The semi-quantitation of lipid droplets was calculated using approximately 500 liver cells.

### 4.5. Immunohistochemistry

The 4 μm thick sections of formalin-fixed tissues were mounted on silanized slides, deparaffinized in xylene, and then rehydrated through a series of alcohol and water baths. We used a Leica DMI-3000 microscope equipped with a digital camera to observe samples, and we used ImageJ (Fiji version 1.8.0) to quantify the staining [17]. The assay was assessed using the Ultravision Quanto Detection system HRP DAB kit (Thermo Scientific Inc., TL-060-QHD; Waltham, MA, USA) by one of the authors (MMT), as well as an independent researcher (YMC).

### 4.6. Terminal Deoxynucleotidyl Transferase-Mediated Deoxyuridine Triphosphate Biotin Nick-End Labeling (TUNEL)

TUNEL assay was performed with fixed tissues embedded in paraffin (4 μm thick sections) and mounted on slides. We used an apoptosis detection kit (Roche, 11684817910, Mannheim, Germany) following the manufacturer’s instructions to assess cellular apoptosis [32]. The rates of TUNEL-positive cells using 10 random fields were calculated by counting the number of positively stained cells in the liver from each rat.

### 4.7. Statistical Analysis

Data analysis was conducted using the Statistical Package for the Social Sciences software Version 24 (SPSS, Chicago, IL, USA). Biochemical parameters, enzyme activities, and Western blot results were analyzed using two-way ANOVA with Tukey post-hoc tests. The comparison of means between two groups was performed using either the Mann–Whitney U test. The results were expressed as mean ± standard error of the mean. A *p*-value of <0.05 was considered statistically significant. Using G*Power 3.1.9.4, the sample size was calculated as four rats per group with the pup’s liver histological semi-quantitative analysis of lipid accumulation. The mean ± standard deviation for NCD group was 1.0 ± 0.63, and for the NCD-H group, it was 8.4 ± 1.01. The sample size calculation aimed to achieve an 80% power to detect a difference between the two groups, using a two-sample *t*-test with a two-sided type I error of 0.05. Based on the calculation, a sample size of three rats per group was determined to be sufficient. However, to ensure a robust sample size, a total of 36 rats were included in the study, with six rats per group.

## 5. Conclusions

The findings of the study indicate that prenatal exposure to microplastics in combination with an HFD resulted in increased hepatic lipid accumulation in the pups. Furthermore, the pups exhibited increased liver apoptosis and inflammation, specifically in response to high-concentration prenatal microplastic exposure with HFD. The study also suggests that oxidative stress and cellular apoptosis play a crucial role in the liver of pups following maternal exposure to microplastics in combination with HFD. The further course of the management study will be focused on anti-inflammation and anti-oxidative stress.

## Figures and Tables

**Figure 1 ijms-24-13457-f001:**
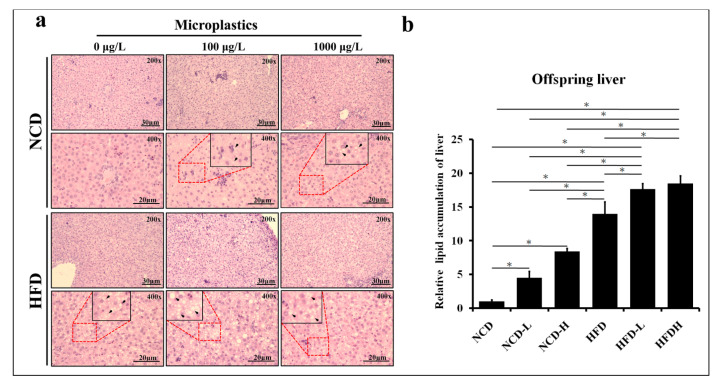
Histological analysis of the liver of pups. (**a**) H&E staining of the liver. (**b**) Semi-quantitative analysis of lipid accumulation. NCD: normal control diet, NCD-L; normal control diet + microplastics (5 μm, 100 μg/L), NCD-H: normal control diet + microplastics (5 μm, 1000 μg/L), HDF: high-fat diet, HFD-L: high-fat diet + microplastics (5 μm, 100 μg/L), HFD-H: high-fat diet + microplastics (5 μm, 1000 μg/L). *: *p* < 0.05. Each group comprised six animals.

**Figure 2 ijms-24-13457-f002:**
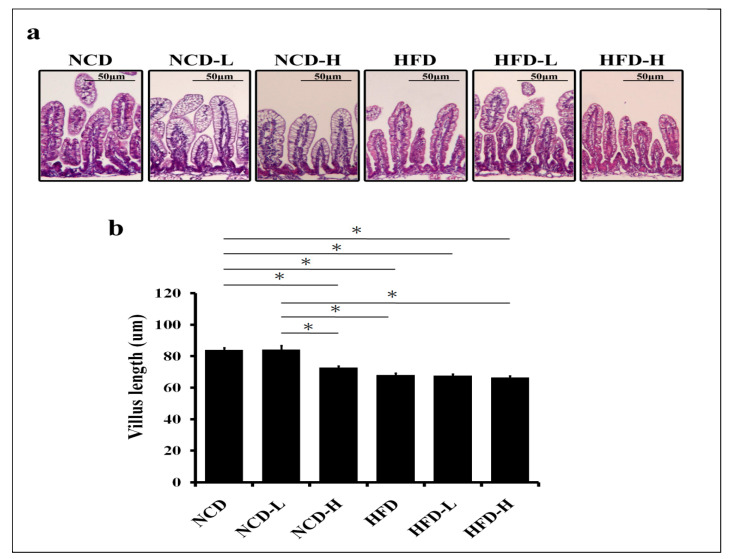
Histological analysis of the ileal tissue of pups. (**a**) H&E staining of the ileal villus length. (**b**) Semi-quantitative analysis of ileal villus length. *: *p* < 0.05. Each group comprised six animals.

**Figure 3 ijms-24-13457-f003:**
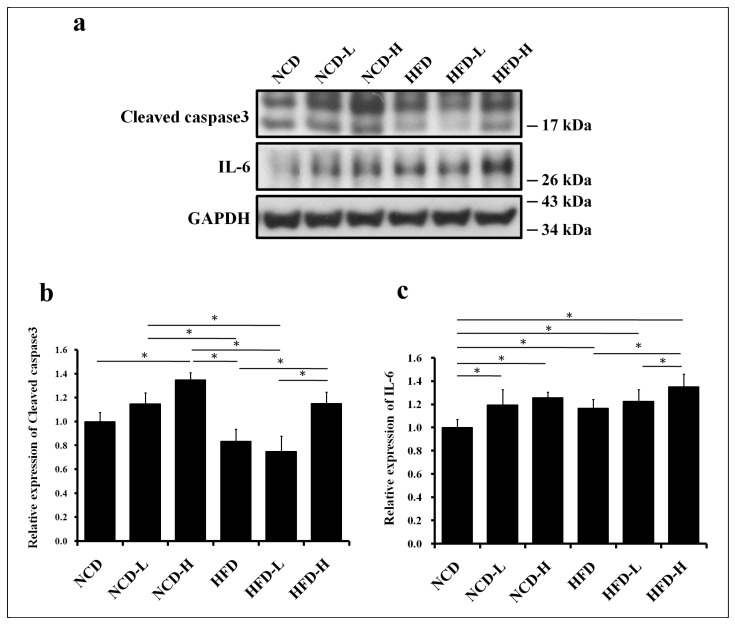
Apoptosis and inflammation in the liver of pups. (**a**) Western blot of caspase-3 and IL-6 expression in the liver of pups. (**b**) Semi-quantitative analysis of liver caspase-3. (**c**) Semi-quantitative analysis of liver IL-6. *: *p* < 0.05. Each group comprised six animals.

**Figure 4 ijms-24-13457-f004:**
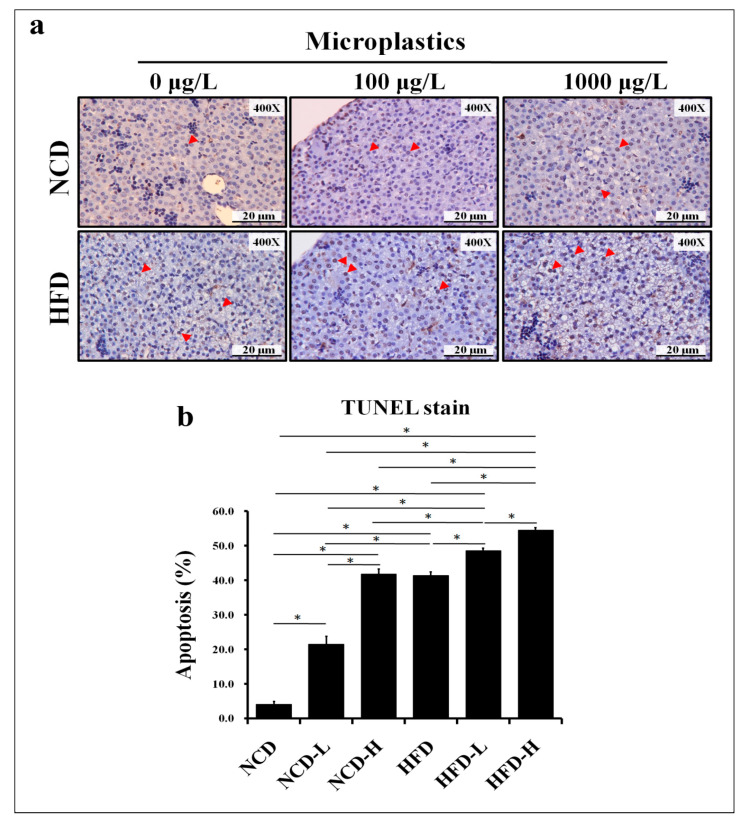
Apoptosis in the liver of pups. (**a**) TUNEL staining of the liver of pups and positive cellular apoptosis staining (triangles). (**b**) Semi-quantitative analysis of liver TUNEL staining. *: *p* < 0.05. Each group comprised six animals.

**Figure 5 ijms-24-13457-f005:**
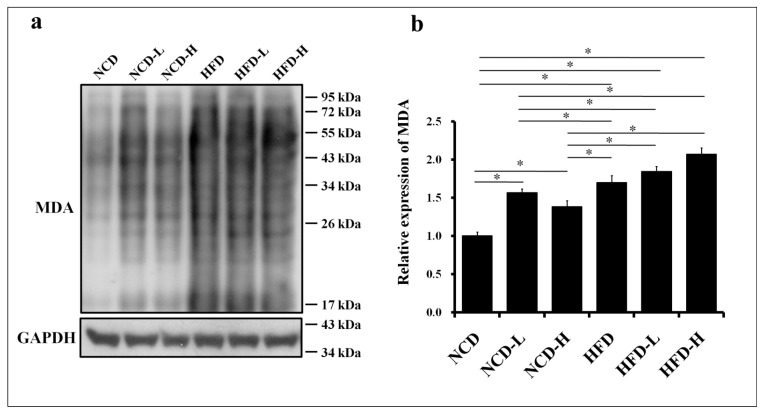
Oxidative stress in the liver of pups. (**a**) Western blot of MDA expression. (**b**) Semi-quantitative analysis of liver MDA. *: *p* < 0.05. Each group comprised six animals.

**Figure 6 ijms-24-13457-f006:**
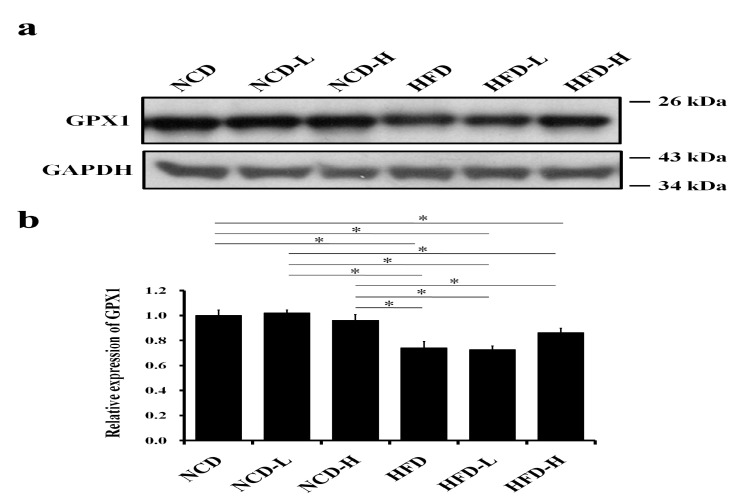
Liver antioxidant level in pups. (**a**) Western blot of Glutathione peroxidase 1 (GPX1) expression. (**b**) Semi-quantitative analysis of liver GPX1. *: *p* < 0.05. Each group comprised six animals.

**Table 1 ijms-24-13457-t001:** Liver weight of mothers, and liver and body weights of pups.

	NCD	NSD-L	NCD-H	HFD	HFD-L	HFD-H
Maternal liver weight (g)	13 ± 1.1	16 ± 0.4	17 ± 0.3	18 ± 4.1 *	18 ± 3.2 *	14 ± 0.5
Pup body weight (g)	9.2 ± 0.3	12.5 ± 0.4 *	13.1 ± 0.5 *	13.3 ± 0.7 *	14.9 ± 0.9 *	15.2 ± 0.5 *
Pup liver weight (g)	0.35 ± 0.03 #$&	0.42 ± 0.03 #$&	0.39 ± 0.05#$&	0.47 ± 0.07 *$	0.57 ± 0.07 *#	0.52 ± 0.06 *

* *p* < 0.05 compared to NCD; # *p* < 0.05 compared to HFD; $ *p* < 0.05 compared to HFD-L; & *p* < 0.05 compared to HFD-H.

## Data Availability

Data from this study will be made available by the corresponding author upon reasonable request.

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
