# Peer review of "Prenatal High-Fat Diet Combined with Microplastic Exposure Induces Liver Injury via Oxidative Stress in Male Pups"

_ijms, 2023, doi:10.3390/ijms241713457_

Round 1

Reviewer 1 Report

The paper entitled: "Prenatal high-fat diet combined with microplastics exposure induces offspring liver injury via oxidative stress" deals with an important topic that is of particular interest in today time. The combined effect of high-fat diet and microplastics on oxidative stress is examined.

The results can serve as a further basis for research on different groups, including future research on the impact on humans.

This reviewer suggests that the paper be accepted after minor revision.

Below are the comments that would improve the quality of this paper:

1. Introduction: I advise that the first two sentences be combined. Rephrase the sentence. Some dietary changes can be positive, and some not.

2. Introduction: I advise you to explain in 1-2 sentences, so far, the most accepted hypothesis of the mechanism of influence of high-fat diet and microplastics on the liver.

3. I agree with the authors who pointed out at the end of the discussion which new results were obtained through this research. I advise that the limitations should be separated as a new section and that the strengths of this research should be added.

4. I advise adding one sentence in the conclusion about recommendations for the further course of the study.

The structure of the work is appropriate.

The methodology of this work is appropriate.

The obtained results are adequately presented in tables and figures.

References are relevant and up-to-date.

Minor language errors are found in the text and are not particularly significant.

Author Response

Response to Reviewer 1:

Thanks for the valuable comments:

  1. Introduction: I advise that the first two sentences be combined. Rephrase the sentence. Some dietary changes can be positive, and some not.

A: Thank you for the comment. We have rephrased the sentence in line 27-43.

  1. Introduction: I advise you to explain in 1-2 sentences, so far, the most accepted hypothesis of the mechanism of influence of high-fat diet and microplastics on the liver.

A: Thank you for the comment. We have added the sentence in line 44-47.

  1. I agree with the authors who pointed out at the end of the discussion which new results were obtained through this research. I advise that the limitations should be separated as a new section and that the strengths of this research should be added.

A: We have separated the section of the limitations and strengths according to line 196-210, respectively.

  1. I advise adding one sentence in the conclusion about recommendations for the further course of the study.

 A: Thank you. We have added the sentence according to line 210-212 and 307-309.

Reviewer 2 Report

This is an interesting manuscript focusing on the relevant clinical question, and bringing results with great contribution, but some issues need to be clarified and changed. The study is in general feasible for publication in this journal. To improve the quality of the manuscript, I would like to suggest the following improvements.

Title - The authors must clarify that only male pups were evaluated

I suggest that the word "offspring" be changed to "pups" when referring to the male pups analyzed in the study.

Abstract: The description of the experimental groups as well as the comparisons with the control group should be better clarified.

Introduction: In general, the introduction is written in a confusing way with no link between the information. I believe that a first paragraph with a better approach to the problem can improve the understanding of the text.

Lines 27 - 34: This paragraph does not feel introductory, it lacks information to introduce the main subject.

Results: The session numbers need to be changed: 2.Results...3 Discussion...

Lines 60-61; 82-83; 97-102; 109-112; 122-124; 130-131: Authors should avoid making suggestions and explanations in the results section. This section should be descriptive only and suggestions reserved for the Discussion session.

The quality of the figures in general needs to be improved. Scales on immunohistochemical images are not clearly visible. The molecular weights of the markers must be provided on the Western blot images.

Fig 5 - Why is the MDA Western blot band "dragged"?

Discussion: Line 137-138 "It was also observed that prenatal HFD and high concentration microplastics exposure resulted in a decrease in the length of the offspring's ileum villus" - In the results section the authors describe that there was no decrease for this group.

Regarding the limitations described by the authors, I believe that an important limitation of the study was not carrying out the evaluation in females, as it would bring very important contributions about the transgenerational effect caused by the ingestion of microplastics. Authors must describe about it.

Materials and Methods: What sample “n” of groups? How was the sample calculation done? Authors must provide these data.

Line 226 - Were males and females fed HFD? Authors should make this information clear in the text.

Line 232 - Did the females mate with healthy males or did they both consume HFD?

Why was the specific period of the 14th day of pregnancy chosen to start ingesting microplastics?

Line 236 - "Some pregnant females" How many? This description of the experimental groups is confusing and it is not clear whether there was a control group without ingestion of microplastics (HFD-L: HFD + microplastics (5μm, 237 100μg/L), HFD-H: HFD + microplastics (5μm, 1000μg/L ), NCD-L: NCD + microplastics 238 (5μm, 100μg/L), and NCD-H: NCD + microplastics (5μm, 1000μg/L). Were the comparisons made only between the groups that ingested microplastics? The results section answers this question, but the methodology needs to be clear.

Line 239 - What happened to the females? Why were they not considered for the study?

What happened to the mothers? Were they sacrificed? Was the maternal liver also processed for analysis?

Line 251 Western blot analysis - What tissue was this analysis performed on? It needs to be included in the methodology.

Line 284 - Statistical analysis: It was not clear how the analysis of the immunohistochemistry data was performed. How was the relationship to “quantify the coloration” given and how was this data processed by the statistical analysis?

Author Response

Response to reviewer 2:

Thank you for the valuable comments. Our responses are as follows.

Point 1: Title - The authors must clarify that only male pups were evaluated

Response 1: We have revised the title as follows: “Prenatal high-fat diet combined with microplastics exposure induces male pups’ liver injury via oxidative stress”

Point 2: I suggest that the word "offspring" be changed to "pups" when referring to the male pups analyzed in the study.

Response 2: Thank you. We have changed the wording from “offspring” to “pups” in the entire manuscript.

Point 3: Abstract: The description of the experimental groups as well as the comparisons with the control group should be better clarified.

Response 3: Thank you. We have clarified according to line 12-13 in the abstract.

Point 4: Introduction: In general, the introduction is written in a confusing way with no link between the information. I believe that a first paragraph with a better approach to the problem can improve the understanding of the text.

Response 4: Thank you. We have rewritten and included the mechanism in line 27-47 of the introduction. We described the importance of maternal HFD and microplastics to pups’ liver steatosis. We have combined the first two sentences to introduce the subject of some dietary changes such as high fat diet or microplastics can be on pups’ liver steatosis and described the possible mechanism studied.

Point 5: Lines 27 - 34: This paragraph does not feel introductory, it lacks information to introduce the main subject.

Response 5: According to line 27-47, we described the importance of maternal HFD and microplastics to pups’ liver steatosis. We have combined the first two sentences to introduce the subject of some dietary changes such as high fat diet or microplastics can be on pups’ liver steatosis and described the possible mechanism studied.

Point 6: Results: The session numbers need to be changed: 2. Results...3 Discussion...

Response 6: Thank you. We have corrected as suggested.

Point 7: Lines 60-61; 82-83; 97-102; 109-112; 122-124; 130-131: Authors should avoid making suggestions and explanations in the results section. This section should be descriptive only and suggestions reserved for the Discussion session.

Response 7: Thank you. We have deleted the suggestions and explanations in the results section according to line 64; 86; 100; 106; 116; 122.

Point 8: The quality of the figures in general needs to be improved. Scales on immunohistochemical images are not clearly visible. The molecular weights of the markers must be provided on the Western blot images.

Response 8: Thank you. We have improved on the figures as attached in the manuscript: immunohistochemical images in Figures 1,2 4; and molecular weights provided on the Western blot images in Figures 3,5,6.

Point 9: Fig 5 - Why is the MDA Western blot band "dragged"?

Response 9: Thank you for the question. Proteins modified by malondialdehyde are recognized and detected, so they are not specific proteins. Therefore, the entire protein lane is usually examined to observe all the proteins present. [https://www.novusbio.com/products/malondialdehyde-antibody-11e3_nbp2-59367, cited on 14th Aug, 2023]

Point 10: Discussion: Line 137-138 "It was also observed that prenatal HFD and high concentration microplastics exposure resulted in a decrease in the length of the offspring's ileum villus" - In the results section the authors describe that there was no decrease for this group.

Response 10: In line 128-130, we have corrected it more clearly as “It was also observed that prenatal HFD and high concentration microplastics exposure resulted in a decrease in the length of the pups' ileum villus compared to NCD.” and in line 86-87 “…a decrease was observed in the 86 HFD-L or HFD-H compared to the NCD group…”.

Point 11: Regarding the limitations described by the authors, I believe that an important limitation of the study was not carrying out the evaluation in females, as it would bring very important contributions about the transgenerational effect caused by the ingestion of microplastics. Authors must describe about it.

Response 11: Thank you. In line 208-210, we have added this important information in the limitations section.

Point 12: Materials and Methods: What sample “n” of groups? How was the sample calculation done? Authors must provide these data.

Response 12: In line 293-300, we have added more information on the sample size calculation: “Using G*Power 3.1.9.4, the sample size was calculated as four rats per group with the pup’s liver histological semi-quantitative analysis of lipid accumulation. The mean ± standard deviation for NCD group was 1.0 ± 0.63, and for the NCD-H group, it was 8.4 ± 1.01. The sample size calculation aimed to achieve an 80% power to detect a difference between the two groups, using a two-sample t-test with a two-sided type I error of 0.05. Based on the calculation, a sample size of three rats per group was determined to be sufficient. However, to ensure a robust sample size, a total of 36 rats were included in the study, with six rats per group.”

Point 13: Line 226 - Were males and females fed HFD? Authors should make this information clear in the text.

Response 13: In line 220-221, we added “One group of females was fed a HFD, and the other female group was fed a normal-chow diet (NCD) … “. And in line 226-228 “The male rats consumed NCD except during mating. Afterwards, the female rats were allowed 24 hours to mate with the male rats while consuming HFD and NCD respectively,…”

Point 14: Line 232 - Did the females mate with healthy males or did they both consume HFD?

Response 14: We have clarified in line 226-228, “The male rats consumed NCD except during mating. Afterwards, the female rats were allowed 24 hours to mate with the male rats while consuming HFD and NCD respectively”

Point 15: Why was the specific period of the 14th day of pregnancy chosen to start ingesting microplastics?

Response 15: We have clarified in line 231-233: “…..the microplastics start ingesting were since conception”

Point 16: Line 236 - "Some pregnant females" How many? This description of the experimental groups is confusing and it is not clear whether there was a control group without ingestion of microplastics (HFD-L: HFD + microplastics (5μm, 237 100μg/L), HFD-H: HFD + microplastics (5μm, 1000μg/L ), NCD-L: NCD + microplastics 238 (5μm, 100μg/L), and NCD-H: NCD + microplastics (5μm, 1000μg/L). Were the comparisons made only between the groups that ingested microplastics? The results section answers this question, but the methodology needs to be clear.

Response 16: Thank you. We have rewritten according to line 231-233: “Few pregnant females in the HFD or NCD paradigms (N=8; low concentration=4, high concentration=4), were further fed with microplastics since conception..”; lines 236-237: “We compared NCD with NCD + microplastics and HFD with HFD + microplastics.”

Point 17: Line 239 - What happened to the females? Why were they not considered for the study?

Response 17: In line 208-210, we have added more information in the limitations. We did not study the female for their more sensitive lipid influenced for their different sex hormone and makes the results of the placental transcriptome in a sex-specific manner. [Maternal high-fat diet sex-specifically alters placental morphology and transcriptome in rats: Assessment by next-generation sequencing. Placenta. 2019;78:44-53.] HF diet significantly altered renal transcriptome with female offspring being more HF-sensitive. [High Fat Diets Sex-Specifically Affect the Renal Transcriptome and Program Obesity, Kidney Injury, and Hypertension in the  Offspring. Nutrients. 2017;9(4):357.]

Point 18: What happened to the mothers? Were they sacrificed? Was the maternal liver also processed for analysis?

Response 18: Yes, they were sacrificed. We did not process maternal liver for analysis but mothers’ liver weight. We found their liver weight only significantly different in the HFD and HFD-L compare to NCD as described in Table 1.

Point 19: Line 251 Western blot analysis - What tissue was this analysis performed on? It needs to be included in the methodology.

Response 19: Thank you. We have added it in the 4.3. Western blot part line 249 “.. the liver tissue Western blot was prepared and examined…”.

Point 20: Line 284 - Statistical analysis: It was not clear how the analysis of the immunohistochemistry data was performed. How was the relationship to “quantify the coloration” given and how was this data processed by the statistical analysis?

Response 20: In line 264-270, we have clarified on analysis used: Hematoxylin and Eosin (H&E) staining; “The image was first converted to an 8-bit gray-scale image, which was black-white inverted so, the lipid droplets would appear black. The black-white inverted image was then applied to an upper threshold of 20 out of 255, of the gray scale to remove inter-hepatocyte structures not indicating lipid droplet features, followed by particle analysis. All particles with the circularity between 0.5 and 1.0, and the diameter between 0.1μm and 50μm were counted. The area fraction (%) occupied was by the counted particles.” The semi-quantitation of lipid droplets was calculated using approximately 500 liver cells” [Journal of Innovative Optical Health Sciences Vol. 11, No. 4 (2018)]

Reviewer 3 Report

The topic of this manuscript is important for the understanding of the pathology of NAFLD. After reading the manuscript, my comments/questions follow below. Please check all sections very carefully.

ABSTRACT

-“Prenatal high-fat diet (HFD) or exposure to microplastics can affect the accumulation of liver fat in the offspring, leading to liver cirrhosis”. The data presented do not support the development of cirrhosis.

- What is the importance of analyzing the ileum in male pups?

RESULTS

-   Results section presents paragraphs repetitive. Please to be rewritten.

-  Table 1: When maternal weight was calculated?

- The authors need to mention the protein molecular weight (kDa) in the figures.

- The quantification of superoxide dismutase (SOD), Interleukin 10 (IL-10), and tumor necrosis factor (TNF) in the liver it would be important to substantiate molecular mechanistic.

DISCUSSION

- Is the percentage of fat chosen ideal to be translated into the consumption of a human diet?

- Can results be comparable depending on the duration of the insult? For example: A shorter intervention time?

MATERIAL AND METHODS

- What was the total number of female rats used in the F0 generation for HFD supplementation and mating? And how many males? How many animals from each litter were analyzed? These numbers are important to show the variability among the rats used in this study.

- The HFD was administered during pregnancy? Did they all get pregnant? Did the use of the hyperlipidemic diet influence the percentage of rats with a positive diagnosis of pregnancy? What about term pregnancy?

- The HFD is it animal or vegetable fat?

- Why did the authors only assess male pups? Could sexual dimorphism exist the liver metabolism? The word offspring is inadequate in the manuscript because only male pups were analyzed.

- How it was calculated the minimal number of animals? Please include this information in the statistical section.

- The Western Blot section needs to be more detailed.

- How the analysis of lipid accumulation was performed using the Image J software?

The manuscript is well designed. Moderate editing of English language required.

Author Response

Response to reviewer 3:

Thanks for the valuable comments. Our responses are as follows.

ABSTRACT

Point 1.-“Prenatal high-fat diet (HFD) or exposure to microplastics can affect the accumulation of liver fat in the offspring, leading to liver cirrhosis”. The data presented do not support the development of cirrhosis.

Response 1: Thank you. We have deleted the “cirrhosis” description in line 10 of the abstract.

Point 2.- What is the importance of analyzing the ileum in male pups?

Response 2: The ileum length is related to the gut-liver axis disorder possibility. [Metformin ameliorates maternal high-fat diet-induced maternal dysbiosis and fetal liver apoptosis. Lipids in health and disease. 2021;20:100.; The gut-liver axis in liver disease: Pathophysiological basis for therapy. J Hepatol. 2020 Mar;72(3):558-577.]

RESULTS

Point 3.-   Results section presents paragraphs repetitive. Please to be rewritten.

Response 3: Thank you. We have deleted the suggestions and explanations in the results section according to line 64; 86; 100; 106; 116; 122.

Point 4.-  Table 1: When maternal weight was calculated?

Response 4: Thank you. We have added “Calculating the maternal liver weight after the female rats gave birth and were sacrificed, the weight increased… “ in line 72-73.

Point 5.- The authors need to mention the protein molecular weight (kDa) in the figures.

Response 5: Thank you. We have improved on the figures as attached in the manuscript: molecular weights provided on the Western blot images in Figures 3,5,6.

Point 6.- The quantification of superoxide dismutase (SOD), Interleukin 10 (IL-10), and tumor necrosis factor (TNF) in the liver it would be important to substantiate molecular mechanistic.

Response 6: Thank you. We agreed it, we did not study IL-10 for insufficient liver specimen, but SOD, TNF-a showed no difference in this postnatal 7 days’ pups as in the following western blot data.

DISCUSSION

Point 7.- Is the percentage of fat chosen ideal to be translated into the consumption of a human diet?

Response 7: Thank you for the question. The human normal fat diet is 17% or so. [Am J Clin Nutr 2005;81:341–54] It is indeed common to use a higher percentage of fat in animal studies to emphasize the effects of a high-fat diet. This is done to simulate the impact of an unhealthy diet on the body and study its effects on various outcomes.                                                                                                             References:

1.Deleterious effects of high-fat diet on perinatal and postweaning periods in adult rat offspring. Clinical nutrition. 2008;27:623-34.

2.Maternal High-Fat Diet Modulates Cnr1 Gene Expression in Male Rat Offspring. Nutrients. 2021 Aug 22;13(8):2885.

3.Resveratrol ameliorates maternal and post-weaning high-fat diet-induced nonalcoholic fatty liver disease via renin-angiotensin system. Lipids in health and disease. 2018;17:178.

4.Metformin ameliorates maternal high-fat diet-induced maternal dysbiosis and fetal liver apoptosis. Lipids in health and disease. 2021;20:100.

Point 8.- Can results be comparable depending on the duration of the insult? For example: A shorter intervention time?

Response 8: We did not study the shorter time, but a study described that PS-MP particles intraperitoneally injected at a dose of 250 μg in a 200 μL saline solution on days 5.5 and 7.5 of gestation, the most abundant immune cells in placenta during the first trimester, the proportion of CD49b+ NK cells in the CD45+ leukocytes decreased distinctly, but no obvious changes were observed with the population of NK cells in the peripheral blood and spleen. [Reproductive Toxicology 106 (2021) 42–50] 

MATERIAL AND METHODS

Point 9.- What was the total number of female rats used in the F0 generation for HFD supplementation and mating? And how many males? How many animals from each litter were analyzed? These numbers are important to show the variability among the rats used in this study.

Response 9: In the F0 generation, a total of 4 male and 4 female rats were used for mating with HFD supplementation, and another 4 male and 4 female rats were used for mating with a normal diet. In our study, each group had 6 male pups, with three of them randomly selected from two different mothers. The detail of the mother and pups were in the followings.

mothers

sacrificed

pups

Pups died

Survived male pups

Survived female pups

NCD-1

2021/7/12

16

4

8

4

NCD-2

2021/7/16

15

0

8

7

NCDL-1

2021/7/14

16

1

7

8

NCDL-2

2022/7/17

15

0

4

11

NCDH-1

2021/7/14

16

0

7

9

NCDH-2

2022/7/18

16

0

6

10

HFD-1

2021/7/13

13

0

9

4

HFD-2

2021/7/13

16

5

5

6

HFDL-1

2021/7/12

14

0

7

7

HFDL-2

2021/7/12

15

0

7

8

HFDH-1

2021/7/12

13

0

4

9

HFDH-2

2022/7/17

11

0

6

5

Point 10.- The HFD was administered during pregnancy? Did they all get pregnant? Did the use of the hyperlipidemic diet influence the percentage of rats with a positive diagnosis of pregnancy? What about term pregnancy?

Response 10: Yes, the HFD was administered during pregnant, and all rats became pregnant regardless of the diet they were on. The use of hyperlipidemic diet did not influence the percentage of rats with a positive diagnosis of pregnancy. They all are term (GA19-21 days). Additionally, the table above describes the death of the pups.

Point 11.- The HFD is it animal or vegetable fat?

Response 11: Thank you. We have added this in 4.1. Animals in line 223-224 “35.8 g/100 g saturated fat (58 kcal% fat) from soybean oil and coconut oil”.

Point 12.- Why did the authors only assess male pups? Could sexual dimorphism exist the liver metabolism? The word offspring is inadequate in the manuscript because only male pups were analyzed.

Response 12: Thank you for the questions. We have added this in limitations in line 208-210. We did not study the female for their more sensitive lipid influenced for their different sex hormone and makes the results of the placental transcriptome in a sex-specific manner. [Maternal high-fat diet sex-specifically alters placental morphology and transcriptome in rats: Assessment by next-generation sequencing. Placenta. 2019;78:44-53.] HF diet significantly altered renal transcriptome with female offspring being more HF-sensitive. [High Fat Diets Sex-Specifically Affect the Renal Transcriptome and Program Obesity, Kidney Injury, and Hypertension in the Offspring. Nutrients. 2017;9(4):357. ]

Thank you. We have replaced the word “offspring” as “pups” in the manuscript.

Point 13.- How it was calculated the minimal number of animals? Please include this information in the statistical section.

Response 13: I n line 293-300, we have added more information on the sample size calculation: “Using G*Power 3.1.9.4, the sample size was calculated as four rats per group with the pup’s liver histological semi-quantitative analysis of lipid accumulation. The mean ± standard deviation for NCD group was 1.0 ± 0.63, and for the NCD-H group, it was 8.4 ± 1.01. The sample size calculation aimed to achieve an 80% power to detect a difference between the two groups, using a two-sample t-test with a two-sided type I error of 0.05. Based on the calculation, a sample size of three rats per group was determined to be sufficient. However, to ensure a robust sample size, a total of 36 rats were included in the study, with six rats per group.”

Point 14.- The Western Blot section needs to be more detailed.

Response 14: Thank you. We have improved on the figures as attached in the manuscript: molecular weights provided on the Western blot images in Figures 3,5,6.

Point 15.- How the analysis of lipid accumulation was performed using the Image J software?

Response 15: In line 264-270, we have clarified on analysis used: Hematoxylin and Eosin (H&E) staining; “The image was first converted to an 8-bit gray-scale image, which was black-white inverted so, the lipid droplets would appear black. The black-white inverted image was then applied to an upper threshold of 20 out of 255, of the gray scale to remove inter-hepatocyte structures not indicating lipid droplet features, followed by particle analysis. All particles with the circularity between 0.5 and 1.0, and the diameter between 0.1μm and 50μm were counted. The area fraction (%) occupied was by the counted particles. The semi-quantitation of lipid droplets was calculated using approximately 500 liver cells.” [Journal of Innovative Optical Health Sciences Vol. 11, No. 4 (2018)]

Point 16.The manuscript is well designed. Moderate editing of English language required.

Response 16: Thank you. We had English edited again.
